# A Study of Plant-Filled Polymer Composites Based on Highly Plasticized Polyvinyl Chloride

**DOI:** 10.3390/polym16111551

**Published:** 2024-05-30

**Authors:** Evgeniia Samuilova, Alina Ponomareva, Vera Sitnikova, Anton Zhilenkov, Olga Kichigina, Mayya Uspenskaya

**Affiliations:** 1The Center for Chemical Engineering, ITMO University, 197101 Saint Petersburg, Russia; samuilova.eo@itmo.ru (E.S.); kresenka@gmail.com (V.S.); mv_uspenskaya@itmo.ru (M.U.); 2Department of Cyber-Physical Systems, Saint Petersburg Marine Technical University, Leninskiy Pr. 101, 198303 Saint Petersburg, Russia; zhilenkovanton@gmail.com (A.Z.); olga1986kichigina@gmail.com (O.K.)

**Keywords:** composite, plastic materials, polyvinyl chloride, plant fillers, fabrication method, film characteristics

## Abstract

To enhance the ecological properties of polyvinyl chloride (PVC) products, the fabrication of PVC-based composites using biofillers with acceptable performance characteristics could be considered. In this work, plant-filled PVC-based composite materials were fabricated and their optical, structural, thermal, and mechanical properties, depending on the nature of the filler, were studied. Spruce flour, birch flour, and rice husk were used as fillers. Optical measurements showed the selected technological parameters, allowing films with a uniform distribution of dispersed plant filler in the polymer matrix to be obtained. Using the plant fillers in PVC films leads to a reduction in strength characteristics; for instance, the tensile strength changed from 18.0 MPa (for pure PVC film) to ~7 MPa (for composites with 20 wt.% of fillers), and to ~5–6.2 MPa (for composites with 40 wt.% of fillers). Thermal investigations showed that the samples with plant fillers could be used at low temperatures without changing their operating characteristics. Thus, plant-filled PVC-based composite materials have a wide operating temperature range, from—65 °C to 150 °C. TGA analysis has demonstrated that the rice husk affected the thermal stability of the composites by increasing their thermal decomposition resistance. The ability to absorb water was observed during the investigation of water absorption of the samples. And the highest degree of water absorption (up to 160 mg/g) was detected for the sample with 40 wt.% of rice husk. In general, plant-filled polymer composites based on PVC can be used on an equal basis with unfilled PVC plastic compounds for some applications such as in construction (for example, for design tasks).

## 1. Introduction

Currently, due to the development of the chemical industry and available technological capabilities, products made from polymer materials are represented in many sectors of consumer goods: construction, automotive industry, electronics, and medicine [1]. These can be polymer membranes for various purposes [2,3], solid-state lithium–metal batteries [4], dielectric materials [5], packaging [6,7], anti-corrosion coatings [8], and construction materials [9,10,11].

The strength, lightness, and other properties of plastic have allowed it to establish a significant presence in the everyday life of modern people. One of the most common types of plastic is polyvinyl chloride (PVC), which is used in practically all spheres of human activity [12,13,14,15,16,17,18]. Plasticized polyvinyl chloride (PVC-P) is one of the most used materials for the manufacture of various products [19,20,21,22,23,24]. The material is made soft and elastic due to the addition of up to 50 wt.% plasticizers. However, in highly plasticized PVC, the process of plasticizer migration occurs while using products made from such a material [25,26,27], which is also an important aspect to study.

One of the significant drawbacks of PVC-based products, which can have a negative impact on the environment, is their long lifespan and inability to biodegrade, so the issue of waste disposal of PVC is quite urgent [28,29,30,31,32,33]. Waste disposal by burning is also not a solution for this problem due to the pollution released into the atmosphere that is harmful to the environment and humans [34,35,36,37]. Thus, it is necessary to consider alternative ways to produce materials and prevent the accumulation of polymers in the environment, for example, the creation of composites containing biopolymers and/or plant fillers. Replacing part of the primary PVC in the composition of the composite with a mineral or vegetable filler will reduce the proportion of PVC in the product and, consequently, its accumulation in the environment. And choosing the correct ratio of PVC and filler will make it possible to obtain materials with acceptable performance characteristics, which will ultimately have a positive impact on the environmental situation in the world [38,39,40,41,42,43]. In addition, the use of agricultural waste as plant fillers, the volume of which in the world annually amounts to about 140 billion tons [44], will also improve the ecology because of the reduction of such waste.

Such fillers do not change the PVC macromolecules themselves; however, they can penetrate between chains of their structure, which leads to a change in the free volume between them. This can happen in different ways depending on the properties of each filler, its structure, and polarity. However, it should be noted that fillers also promote the formation of dense packing of PVC powder due to the interaction between the PVC particle and the additive. As a result, an interparticle interaction is reduced [45].

One of the well-known types of PVC-based composites is wood–polymer or plant-filled composites, which, in addition to PVC and filler, may contain additional components to control the properties of the final product [46,47,48,49,50,51,52]. Wood–polymer composites have been known since the beginning of the 20th century [53]. At first, they were used in the production of decking, gradually expanding their scope of application [54]. Besides wood components, composites can include various plant fillers, such as bamboo, wheat straw, and fruit fibers [52,55,56]. Since the plant filler is a hydrophilic and combustible material, developments are underway not only for the determination of the optimal ratio of PVC and plant filler, but also for enhancing thermal and mechanical characteristics of wood-polymer composites as well as for finding pre-treatment technologies in order to improve performance characteristics [51,52,57,58,59]. The effect of filler on the processability of a mixture of PVC and wood components is also under investigation [60].

However, most researchers of plant-filled PVC composites focus on the fabrication of 3D products (decking, bars, panels). Plant-filled composites based on highly plasticized PVC in the form of films have rarely been studied. Such materials could be used in the production of flooring, fancy goods, or packaging. The aim of this work was to obtain plant-filled polymer composites based on highly plasticized polyvinyl chloride filled with various plant fillers and to study their properties depending on the nature of the filler.

## 2. Materials and Methods

### 2.1. Polymer Compositions

The objects of study were polymer films based on an industrial composition of highly plasticized PVC (Klöckner Pentaplast Rus Ltd., Saint Petersburg, Russia), containing various plant fillers with a weight composition of 0–40 wt.%. The industrial composition of highly plasticized PVC (Klöckner Pentaplast Rus Ltd.) contains suspension polyvinyl chloride (RusVinyl Ltd., Nizhny Novgorod, Russia) with the Fickentcher constant (K_f_) equal to 67 and plasticizer dioctyl terephthalate Eastman 168 (Eastman, Kingsport, TN, USA). Dioctyltin bis(2-ethylhexyl thioglycolate) (Galata Chemicals, Southbury, CT, USA) with a concentration of 1 wt.% as a thermal stabilizer in the industrial PVC composition, glycerol ether (Emery Oleochemicals, Telok Panglima Garang, Malaysia) as an internal lubricant of 0.5 wt.%, and oxidized polyethylene wax (Honeywell, Charlotte, NC, USA) as an external lubricant of 0.1 wt.% were used.

Spruce flour (SF) (Slotex Ltd., Saint Petersburg, Russia), lignin-free birch flour (BF) (Lignum-Resource Ltd., Kazan, Russia), and rice husk (RH) (SkifAgros Ltd., Zernograd, Russia) were chosen as plant fillers. SF is residue obtained during the processing of silver spruce (*Abies alba*). BF is lignin-free wood flour obtained from waste of silver birch (*Betula pendula Roth*). RH is rice husk variety “Sonata” (*Oryza sativa* L.). All fillers are waste products from the woodworking or agricultural industries. The choice of such components is because they have different strength characteristics and resinous substance content. Table 1 shows the comparison of main parameters of fillers [61].

### 2.2. Polymer Films Fabrication Method

Plant-filled polymer composites were obtained with different ratios of components. Plant filler (PF) was ground previously for five minutes using a roller mill (Brabender, Duisburg, Germany) to obtain a powder with a dispersion of 0.2–2 mm and dried up to 3% humidity. The required fraction of the plant filler was selected by the sieve method. The hygroscopic water content was determined by the gravimetric method.

The components (PVC composition and PF) were mixed for three minutes using a high-speed mixer (HENSCHEL, Kassel, Germany) until a homogeneous mixture was obtained. The resulting mixture was fed to laboratory rollers (Schwabenthan Maschinenfabrik. Berlin, Germany) type Polymix 150U heated to a temperature of 175 °C to obtain a melt. The melt was rolled at the speed of 24 rpm for two minutes to obtain polymer composite films with a thickness of 500 ± 5 µm. The technological scheme is shown in Figure 1.

The containment of filler was 20 or 40 wt.%. The compositions of the films obtained are presented in Table 2.

### 2.3. Investigation Methods

Structural, thermal, and mechanical properties of composite film samples were investigated using different methods. 

#### 2.3.1. Optical Microscopy

Optical microscopy was carried out using an Olympus STM6 (Olympus Corporation, Tokyo, Japan) optical measuring microscope using the transmission method. 

#### 2.3.2. FTIR Spectroscopy

The chemical structure of the samples was studied by Fourier transform infrared spectroscopy. The IR spectra of the samples were recorded on a Bruker Tensor 37 FTIR spectrometer using a Pike MIRacle ATR (Pike Technologies Inc., Madison, WI, USA) (attenuated total internal reflection) attachment with a diamond-coated ZnSe crystal. The spectra were recorded in the range of 4000–600 cm^−1^ with a resolution of 2 cm^−1^ and averaging over 32 spectra.

#### 2.3.3. DSC

The thermal properties of the obtained samples were studied on a differential scanning calorimeter DSC 204 F1 Phoenix (Netzsch, Selb, Germany), operating at a heating rate of 10 °C/min in the range from −30 °C to 175 °C. After an equilibration phase at 25 °C, the samples were first heated, after that cooled to −30 °C, and then heated again to a temperature of 250 °C. Nitrogen was used as the purge gas at a flow rate of 50 mL/min. The weight of the film samples used for DSC measurements varied from 5.5 to 8.5 mg. The samples were placed in aluminum crucibles with a pierced lid. To determine the glass transition temperature T_g_, the data from the second heating curve were analyzed.

#### 2.3.4. TGA

Thermogravimetric analysis was used to study the thermal stability of the samples. All samples were measured using a Libra 209 F1 thermogravimetric analyzer (Netzsch, Selb, Germany) with a scanning speed of 10 °C/min under a nitrogen atmosphere in the temperature range of 25–900 °C. Nitrogen was supplied at a flow rate of 50 mL/min.

#### 2.3.5. Tensile Test

Mechanical tests were conducted in accordance with ISO 527-3:2018 [62] at a strain rate of 180 mm/min using an Instron 5966 electromechanical testing machine (manufactured in the USA) equipped with a 10 kN load cell and a pneumatic gripper. Signal processing was performed using Bluehill 3 software. For the tests, 5 samples of each film composition were prepared with a size of 10 × 50 mm. To avoid significant variations in the results for samples containing plant fillers due to their anisotropic nature, the tests were performed along the direction of the fiber orientation corresponding to the rolling direction. Standard treatment was performed on 5 measurements at a confidence level of 0.9, the standard deviation was calculated and outliers that were not within the 1.5 interquartile range were removed from the analysis.

#### 2.3.6. Water Absorption Test

The water absorption test was conducted according to the methods described in ISO 62:2008 [63] using distilled water at room temperature. The samples were placed in 100 mL of distilled water for a week. Analytical scales (CPA64, Sartorius, Göttingen, Germany) were used to measure the mass of each sample before and after testing. The degree of water absorption (*W*), which is an indicator of the absorption process, was calculated using the following formula: *W* = (*m* − *m*_0_)/*m*_0_ × 100%, where *m* and *m*_0_ are the masses of the sample before and after testing, respectively.

## 3. Results and Discussion

### 3.1. Optical Measurements

When fabricating plant-filled polymer materials, it is especially important to achieve uniform distribution of the filler inside the polymer matrix. An uneven distribution negatively affects the performance characteristics of the materials causing anisotropy of properties and high defectiveness of samples, due to which the formation of products of various configurations from such samples is impossible. 

Images of plant-filled polymer films based on highly plasticized PVC taken in reflected and transmitted light modes are presented in Figure 2 and Figure 3, respectively. As can be seen from the figures, with the selected technological parameters, it is possible to obtain films with a uniform distribution of dispersed plant filler in the polymer matrix. It has also been observed that when the melt is rolled, the particles of the plant filler tend to align themselves mainly along the direction of rolling. The uniform distribution of dispersed plant filler in the polymer matrix is shown even when the proportion of plant filler increases from 20 wt.% to 40 wt.%, as evidenced by the uniformity of the color of the films and the absence of clusters of plant filler. The effect obtained is probably achieved due to the stage of mixing the dry PVC composition and filler before rolling. There is one more feature of the films, which is that all fillers gave their own color to the samples. This can be used to save coloring agents during production.

### 3.2. IR Spectroscopy

IR spectroscopy data for plant-filled polymer films based on highly plasticized PVC are presented in Figure 4. As can be seen from the data obtained, a peak with a wave number of 832 cm^−1^ and 1018 cm^−1^ corresponding to vibrations of the C–Cl bond is identified in the IR spectra of all samples. This peak is characteristic of PVC. In the wavenumber range of 2950–2900 cm^−1^, there are peaks corresponding to the components of the plant filler. The peak at 3336 cm^−1^ corresponds to the O–H stretch. Absorption bands at 2850 cm^−1^ and 2873 cm^−1^ correspond to the C–H stretch in CH_3_ and CH_2_ groups [64]. The peaks around 1400 cm^−1^ are assigned to the C–H aliphatic bending bond. The peak at 1250 cm^−1^ is attributed to the bending bond of C–H near Cl. The C–C stretching bond of the PVC backbone chain occurs in the range 1000–1100 cm^−1^. Finally, peaks in the range of 600–650 cm^−1^ correspond to the C–Cl gauche bond [65]. The peak at 1649 is associated with the absorbed O–H and conjugated C–O bonds.

### 3.3. Thermal Properties

Since plant raw materials are sensitive to heat treatment and are characterized by flammability and poor thermal stability, the study of the thermal characteristics of the composites obtained in the work is important to determine the conditions of applicability of such polymeric materials. Thermograms were carried out by differential scanning calorimetry (Figure 5). All DSC curves show the absence of a glass transition temperature for PVC. This is primarily due to the presence of a high-molecular plasticizer in the composition of the composites, which is dioctyl terephthalate with a glass transition temperature of minus 67 °C.

Plant-filled polymer films based on highly plasticized PVC were investigated by thermogravimetric analysis under argon atmosphere to understand their thermal behavior. Thermograms of the studied polymer composite films are shown in Figure 6.

Typically, two stages of degradation are observed in PVC: the first stage occurred at 250–350 °C and is the destructive processes of PVC associated with the dehydrochlorination reaction and the removal of some volatile small molecules of saturated and unsaturated aliphatic or aromatic hydrocarbons [66]. The second stage occurred in the temperature range of 350–500 °C and includes the formation and volatilization of intramolecular cyclization products [66,67].

Table 3 shows the characterized temperature values for the decomposition stages of samples. As can be seen from the presented data in Figure 6 and in Table 3, plant-filled polymer films based on highly plasticized PVC are also characterized mostly by a two-stage destruction process that is in good agreement with data obtained from other studies [58]. The addition of birch and spruce flour as filler in the polymer composition leads to a slight decrease in the initial decomposition temperature (T_d_) of the material by 10–15 °C. In the case of adding RH filler in the initial polymer mixture, an increase in the maximum temperature of decomposition in the first stage is observed, which is probably due to the RH thermal behavior in the range of 250−350 °C. RH has a different decomposition maximum temperature (~303 °C [68]) in comparison with PVC (~280 °C). For all composite samples, the widening of thermal ranges for both stages of decomposition can be noted. Moreover, after 550 °C the mass remained stable and the residual ash corresponded to 8–10% for SF-filled, to 5.6–13% BF-filled, and to 8.9–10% polymer films of the total weight that was measured.

### 3.4. Mechanical Properties

The results of the tensile mechanical performance tests on the plant-filled polymer films based on highly plasticized PVC are presented in Table 4 and Figure 7. 

All plant-filled PVC composites exhibit the same pattern (Table 4). The addition of the filler in the composite led to a decrease in the strength properties of the material, since filler particles filled the intermolecular space, breaking the integrity and bonds of the polymer fibers. Despite the fact that there are many studies showing that a plant filler improves mechanical characteristics due to the reinforcing effect [69,70,71], our data are in good agreement with the data obtained by other researchers in [72,73,74,75,76,77,78,79]. The more filler, the denser the structure of the material, which means less space is left for PVC molecules to straighten. As a result, the material becomes less flexible, and the value of breaking elongation decreases. Comparing the characteristics of composites with various fillers, it can be noted that the most preferable filler is BF (20%). Thus, increasing the proportion of plant filler to improve the environmental friendliness of materials is limited by decreasing tensile strength characteristics and flexibility.

Since additional processing of plant fibers to increase their adhesiveness was not carried out, a decrease in the strength characteristics of plant-filled polymer films based on highly plasticized PVC was expected. This could be explained by the ineffective transfer of stress from the polymer matrix to the plant fibers due to the poor interaction of the plant fiber with the polymer matrix caused by the different chemical natures of the polymer matrix and fiber reinforcement [72,80]. Also, differences in mechanical characteristics may be associated with different dispersions of the plant filler, which is also noted in the literature [81].

### 3.5. Water Absorption Test

Studying the stability of composite materials based on polymer materials filled with a hydrophilic filler, such as plant fibers, is an important part of the research since water absorption can lead to negative phenomena, such as a decrease in strength, deformation of the product until its complete destruction, oxidation, etc. So, in this work, the effect of water absorption on the resulting plant-filled polymer films based on highly plasticized PVC with various plant additives was studied.

Table 5 presents data on the degree of water absorption of some samples of the obtained plant-filled polymer films based on highly plasticized PVC in water at a temperature of 25 °C for 24 h and 7 days according to the standard technique [63]. And Figure 8 shows the dynamic process of water absorption.

As can be seen from the table, the higher the content of hydrophilic filler, the higher the degree of water absorption of the filled polymer films. This is due to the fact that with an increase in the content of plant filler, the number of free OH groups of cellulose also increases, which, in turn, makes the composites more hydrophilic. These free OH groups interact with water molecules to form a hydrogen bond, which leads to an increase in the weight of the composites [82,83]. As can be seen from the water absorption curves, the samples containing rice husk as a plant filler have the maximum degree of absorption among all fillers This may be explained by the higher ratio of cellulose to lignin than in the other fillers considered (Table 1). Composites containing birch flour have a higher degree of absorption than composites filled with spruce flour, which is easily explained by the removal of lignin, which has hydrophobic properties, from birch flour. In the previous studies conducted by Fedele and Ata [84] and Li et al. [85], it was noted that lignin increases the hydrophobicity of films due to the presence of aromatic functional groups, which can cross-link with polysaccharides and reduce the ability to absorb water.

The degree of water absorption of the samples at a temperature of 25 °C does not exceed 0.161 g/g, which indicates that these materials can be classified as hydrophilic. This may be due to the dispersed structure of the obtained composite as well as the effects of diffusion and percolation. Thus, in [83], three mechanisms were identified by which composite materials can absorb moisture: (1) diffusion of water molecules inside microgaps between polymer chains, (2) capillary transport of water molecules into gaps and defects at the fiber–matrix interface, and (3) transport of water through the microcracks of the matrix formed during the fabrication of the composite. It also explains the increase in the degree of water absorption with an increase in the proportion of plant filler.

## 4. Conclusions

In this work, the properties of PVC-based composite materials with proportions of different types of fillers up to 40% were analyzed. Increasing the amount of plant filler in PVC composite films to improve the environmental friendliness of materials somewhat reduces the strength characteristics and reduces the flexibility of the samples. 

In general, the analysis of the properties of PVC-based composite films has shown the following results:The relatively low glass transition temperature (below—65 °C) of samples with plant filler allows them to maintain their flexibility and elasticity at negative temperatures, which means such composites can be used in a wide range of temperatures and operating conditions.During mechanical tests, it was found that the addition of a plant filler in the amount of 20 wt.% reduces the tensile strength by more than two times, for example, from 18.0 MPa to 7.4 MPa when adding the spruce flour. Increasing the proportion of plant filler in the composites to 40 wt.% leads to a further decrease in tensile strength to 5–6 MPa, depending on the type of filler. A similar situation was observed for elongation strength values. A decrease in elongation at break values by 4–6 times was noted, depending on the type and proportion of filler. To minimize this effect, it is necessary to treat the plant filler with various modifiers.The investigation of water absorption of the samples demonstrated the ability of the PVC- filled composites to absorb water. And the highest degree of water absorption was detected for 60/40-RH (up to 160 mg/g). So, these samples cannot be used in conditions of constant contact with water due to their limited hydrophobicity, but they can withstand repeated short-term exposure to water without loss of performance characteristics.

Thus, plant-filled polymer composites based on PVC can be used instead of unfilled PVC plastic compounds for some applications. Although the reduction in performance characteristics (mechanical, thermal) is significant and limits the use of such materials, they remain suitable for use in areas where mechanical strength is not required, for example, in the production of decorative finishing panels, stationery, and packaging. Moreover, each type of filler can give its own color to the product, which is varied by its proportion. This could save on dyes, some of which are expensive and environmentally hazardous.

## Figures and Tables

**Figure 1 polymers-16-01551-f001:**
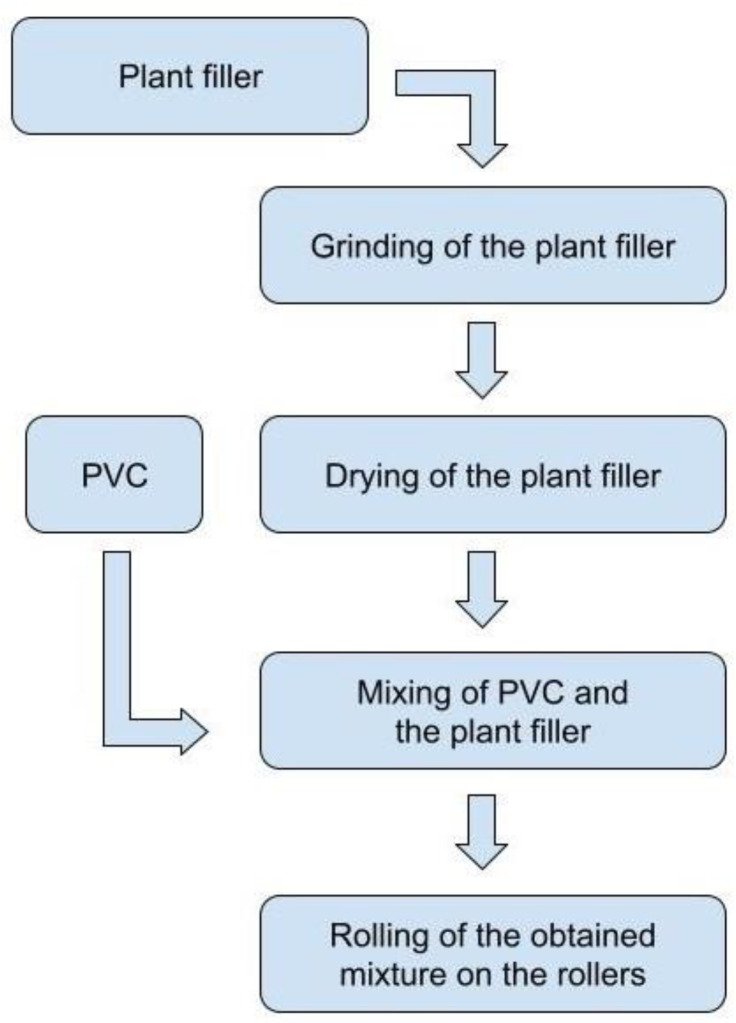
Scheme for the fabrication of plant-filled polymer films based on highly plasticized PVC.

**Figure 2 polymers-16-01551-f002:**
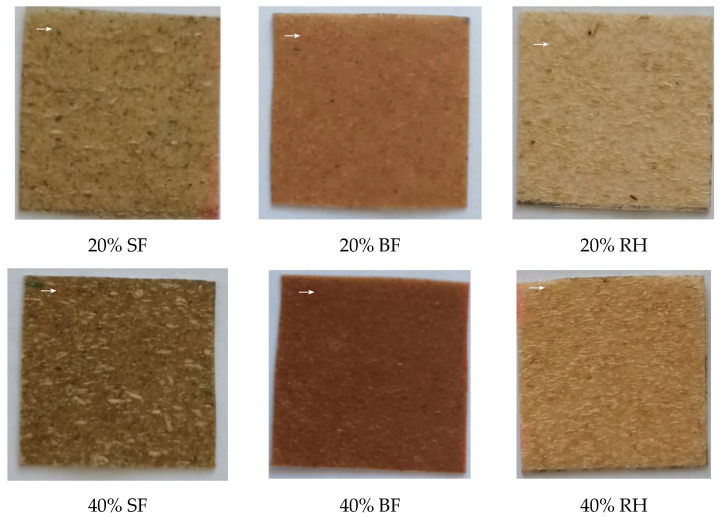
Images of plant-filled polymer films based on highly plasticized PVC taken in the reflected light mode (sample size of 30 × 30 mm). The arrow indicates the predominant orientation direction of the filler.

**Figure 3 polymers-16-01551-f003:**
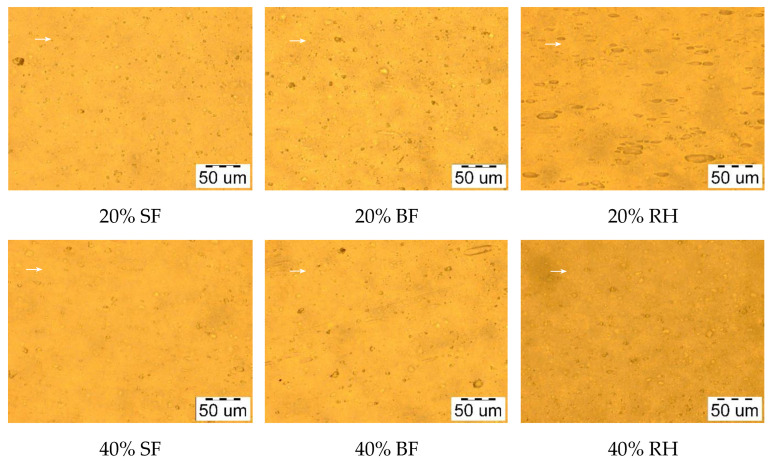
Images of plant-filled polymer films based on highly plasticized PVC taken in the transmitted light mode (sample size of 30 × 30 mm). The arrow indicates the predominant orientation direction of the filler.

**Figure 4 polymers-16-01551-f004:**
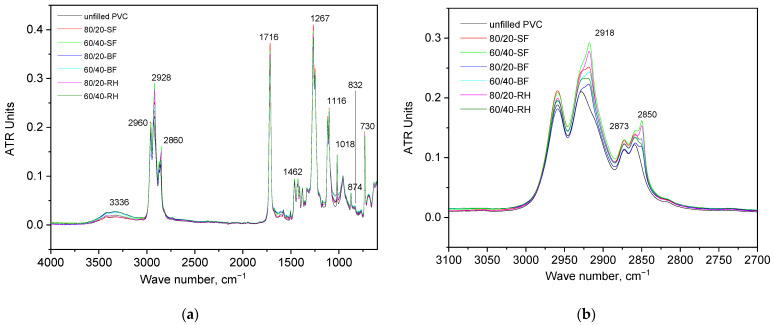
Averaged IR spectra of plant-filled polymer films based on highly plasticized PVC: (**a**) full spectra range and (**b**) the characterized peaks of plant fillers, where 1—unfilled PVC, 2—80/20−SF, 3—60/40−SF, 4—80/20−BF, 5—60/40-BF, 6—80/20−RH, 7—60/40−RH.

**Figure 5 polymers-16-01551-f005:**
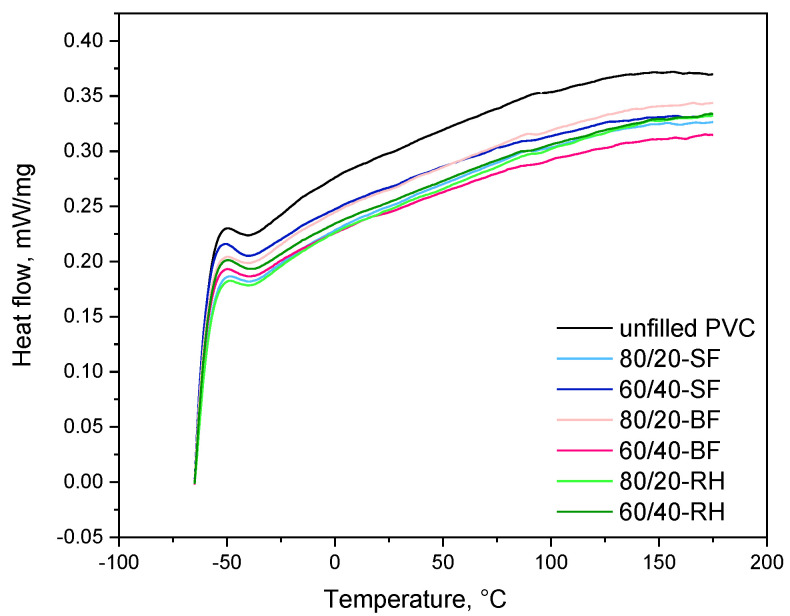
DSC of plant-filled polymer films based on highly plasticized PVC: 1—PVC, 2—80/20-SF, 3—60/40−SF, 4—80/20−BF, 5—60/40−BF, 6—80/20−RH, 7—60/40−RH.

**Figure 6 polymers-16-01551-f006:**
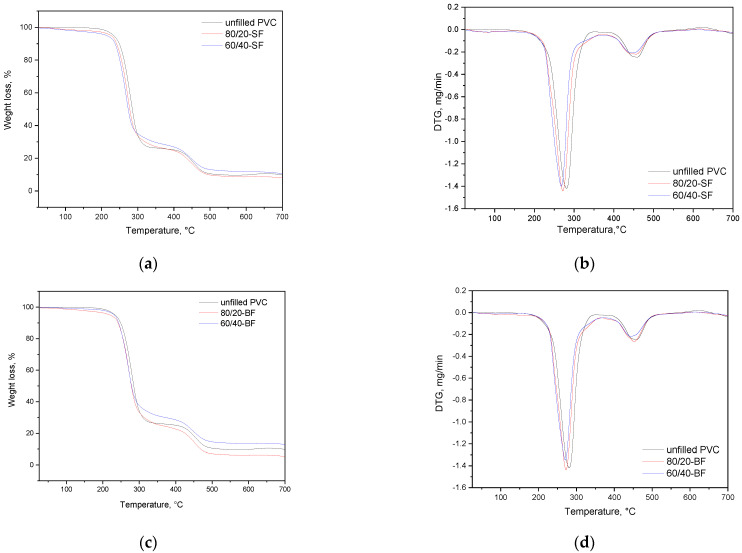
TGA and DTG curves of plant-filled polymer films based on highly plasticized PVC: (**a**,**b**)—with addition of SF, (**c**,**d**)—with addition of BF, (**e**,**f**)—with addition of RH.

**Figure 7 polymers-16-01551-f007:**
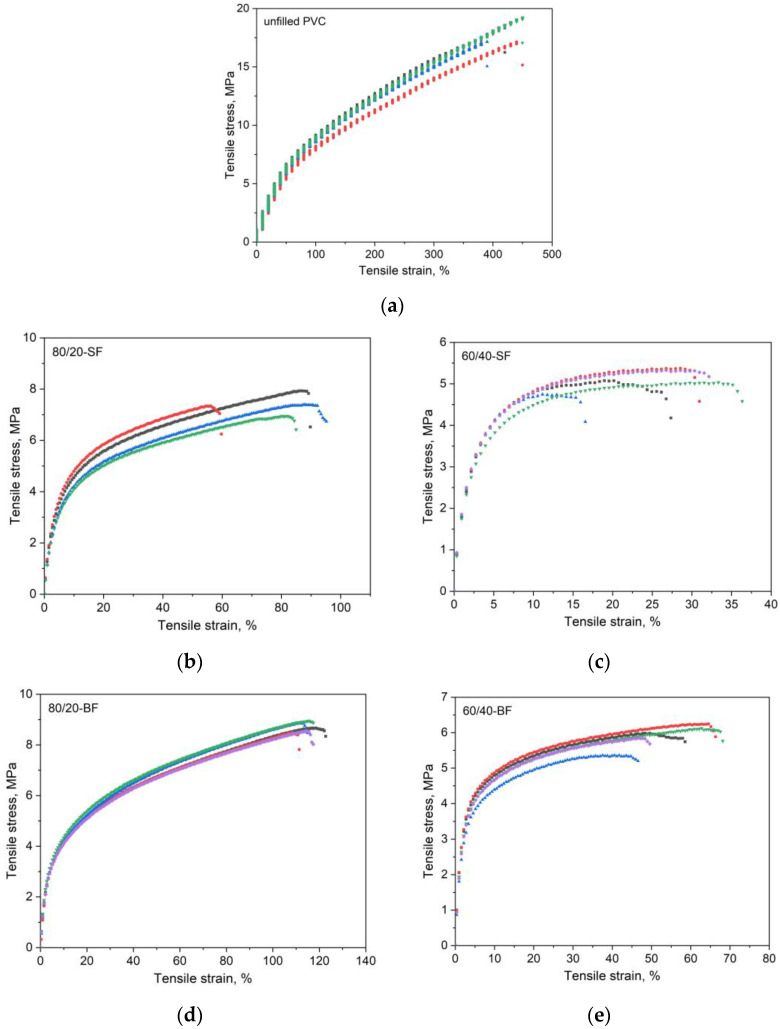
Stress–strain diagrams of plant-filled polymer films based on highly plasticized PVC, where (**a**)—unfilled PVC, (**b**)—80/20-SF, (**c**)—60/40-SF, (**d**)—80/20-BF, (**e**)—60/40-BF, (**f**)—80/20-RH, (**g**)—60/40-RH. Colors show the different strain-stress measurements of the sample (from 1 to 5).

**Figure 8 polymers-16-01551-f008:**
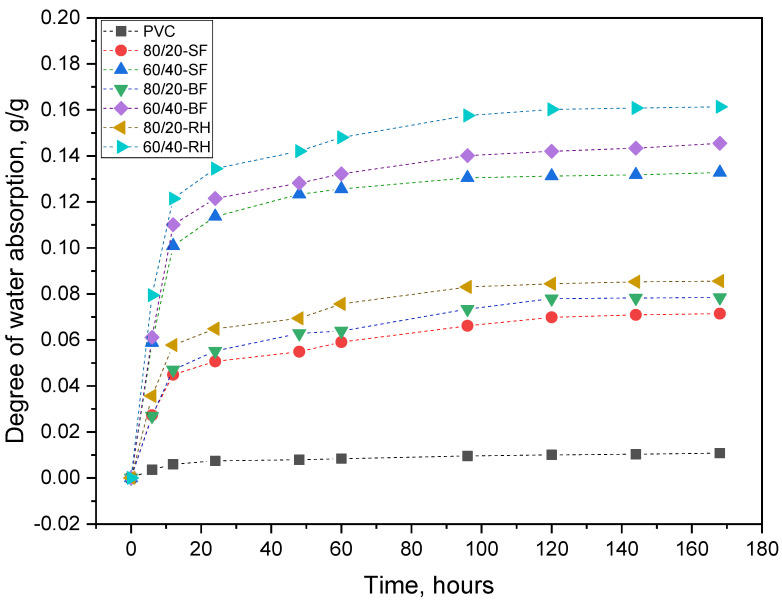
The degree of water absorption of plant-filled polymer films based on highly plasticized PVC for 7 days.

**Table 1 polymers-16-01551-t001:** Comparison of the plant fillers used for film fabrications.

Parameter	Plant Filler		
	SF	BF	RH
Brinell hardness, kgf/mm^2^	2.5	3	-
Yankee hardness, ft	660	1260	-
Density, kg/m^3^	400–500	540–700	735
Tensile strength along/across the fibers, MPa	98/4.7	158/10.6	-
Moisture stability	medium-stable **	relatively stable *	stable ***
Organic substances, % [53]: cellulose hemicellulose lignin ash content			
	42 ± 2	44 ± 3	38 ± 8
	26 ± 3	32 ± 5	25 ± 3
	29 ± 4	18 ± 4	14 ± 2
	0.2–0.8	0.2–0.8	19
cellulose/lignin	1.45	2.44	2.71

*—very slightly deformed with small changes in air humidity; **—visibly deformed with small changes in air humidity; ***—practically not deformed with small changes in air humidity.

**Table 2 polymers-16-01551-t002:** Compositions of the studied plant-filled polymer films based on highly plasticized PVC.

Composition of Samples PVC/Plant Filler	Component, wt.%			
	PVC Composition	Plant Filler		
		Spruce Flour	Birch Flour	Rice Husk
PVC	100	-	-	-
80/20-SF	80	20	-	-
60/40-SF	60	40	-	-
80/20-BF	80	-	20	-
60/40-BF	60	-	40	-
80/20-RH	80	-	-	20
60/40-RH	60	-	-	40

**Table 3 polymers-16-01551-t003:** Results of thermogravimetric analysis of plant-filled polymer films based on highly plasticized PVC.

Sample	Temperature Value at Mass Loss during the Decomposition (°C)			
	T_max_ of 1st Stage	1st Stage Range	T_max_ of 2nd Stage	2nd Stage Range
PVC	282	246–347	462	425–558
80/20-SF	273	240–368	458	418–557
60/40-SF	267	235–362	447	416–562
80/20-BF	272	244–366	452	424–567
60/40-BF	269	242–372	444	415–563
80/20-RH	297	246–369	454	414–568
60/40-RH	297	244–367	443	414–565

**Table 4 polymers-16-01551-t004:** Mechanical characteristics of plant-filled polymer films based on highly plasticized PVC.

Sample	Thickness of the Samples, mm	Tensile Strength, MPa	Relative Elongation at Break,%	Young’s Modulus, MPa
PVC	0.594	18.0 ± 1.0	427.4 ± 29.3	14.5 ± 0.7
80/20-SF	0.501	7.4 ± 0.4	124.7 ± 28.2	94.9 ± 9.1
60/40-SF	0.511	5.1 ± 0.2	62.4 ± 7.6	148.2 ± 4.7
80/20-BF	0.523	8.7 ± 0.2	140.0 ± 12.5	96.8 ± 8.2
60/40-BF	0.483	5.9 ± 0.3	131.8 ± 11.1	162.8 ± 9.7
80/20-RH	0.520	6.9 ± 0.8	126.9 ± 20.4	120.4 ± 10.6
60/40-RH	0.543	6.2 ± 0.2	81.6 ± 11.1	206.6 ± 4.9

**Table 5 polymers-16-01551-t005:** The degree of water absorption of plant-filled polymer films based on highly plasticized PVC.

Sample	Water Absorption Degree, g/g	
	After 24 h	After 7 Days
PVC	0.007 ± 0.002	0.011 ± 0.003
80/20-SF	0.051 ± 0.018	0.071 ± 0.009
60/40-SF	0.114 ± 0.011	0.133 ± 0.006
80/20-BF	0.055 ± 0.007	0.078 ± 0.011
60/40-BF	0.121 ± 0.015	0.145 ± 0.017
80/20-RH	0.065 ± 0.009	0.086 ± 0.014
60/40-RH	0.134 ± 0.011	0.161 ± 0.007

## Data Availability

The original contributions presented in the study are included in the article material, further inquiries can be directed to the corresponding author.

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
