# Peer review of "A Study of Plant-Filled Polymer Composites Based on Highly Plasticized Polyvinyl Chloride"

_polymers, 2024, doi:10.3390/polym16111551_

Round 1

Reviewer 1 Report

Comments and Suggestions for Authors

Plant-filled PVC-based composites were obtained and their optical, structural, thermal, and mechanical properties depending on the nature of the filler (spruce flour, birch flour, and rice husk) were studied. Increasing the proportion of plant filler somewhat reduces the strength characteristics and reduces the flexibility of the samples. But samples with plant filler allows them to be used at low temperatures without changing the operating characteristics. I believe this paper can be published in the journal if the authors address the above comments effectively and revise their manuscript.

1- English of the manuscript needs polishing.

2- The Abstract in its current form is not sufficient. In particular, it should be supported in a more effective manner by the results obtained during research, because the first part which is read by journal's audience is Abstract and thus it should reflect the novelty and perform the main results.

3- Please add more information about plant fillers?

4- The Introduction Section in its current form is not adequate. In this context, I strongly recommend the author(s) to analyze and discuss separately each cited paper. Besides, the differences/advantages of the present investigation compared to other literature works should be written out at the end of this Section in a much more detailed and comprehensive manner.

5- More physical interpretation about the results can improve the quality of this work.

6- Is it possible to add the results related to the elastic modulus for the studied composites?

Comments on the Quality of English Language

English of the manuscript needs polishing.

Author Response

Answers in the report, attached.

Reviewer 2 Report

Comments and Suggestions for Authors

This manuscript presents the effects of filler type and content on the physical, thermal, mechanical and thickness swelling properties of PVC-based composites. Although the material properties of green materials are worth investigating, the value of publishing the findings will be lost if the manuscript is not written properly. The authors intended to resolve the non-biodegradability of PVC polymer by incorporating biofillers. However, incorporating biofillers actually could not promote the biodegradability of PVC polymer. As such, I could not see the motivation for conducting this research study. In the introduction, the authors also did not emphasize and state the novelty/innovative aspects of this work very clearly. In addition, this research study aims to identify the effect of filler content and filler types on the physical, thermal, mechanical and thickness swelling properties of PVC-based composites, but no results were analyzed, nor was an in-depth discussion on these topics made. Based on my assessment, this manuscript lacks sufficient knowledge depth and does not attain the desirable level of publishable standard, particularly for the journal Polymers. This manuscript can only be reconsidered if the authors could substantially improve the manuscript quality and provide an in-depth discussion on the subject matter.

1. The authors must seek an English language editor to improve the language quality. Many sentences are hard to understand.

2. The findings, as stated in the abstract, are too general. Please add major findings with some qualitative results.

3. On page 1; line 40, the long lifespan of the materials is not a disadvantage. Instead, the main disadvantage of the polymer is its non-biodegradability.

4. On page 2; lines 50 – 51, the ‘volume of external voids between them’ is not understandable. Please revise the sentence.

5. The introduction needs a major revision. The objective of this work does not match the problem statement. Incorporating biofillers does not promote the biodegradation of PVC polymer. Besides, the introduction is supposed to give a better picture of what is lacking in the literature studies (research gap) and state the novelty of this work very clearly. However, these aspects have been omitted from the manuscript.

6. On page 2; lines 64 – 65, please change ‘concentration’ to ‘weight composition’.

7. In section 2.1, please provide the source/supplier of each material used in this work.

8. On page 2; line 79, how did the authors measure the humidity of 3 %?

9. On page 3; lines 109 – 113, what is the difference between tensile strain rate and loading speed? Why was the loading speed higher than what was generally fixed in other literature studies? Did the authors refer to any standards when performing the tensile tests?

10. On page 7; line 184, how did the authors perform stretching on the sample? It should be described in the methodology section.

11. For section 3.4, the authors should provide the stress-strain curves of the composites. Besides, the tensile modulus of the materials should be added to Table 3.

12. When looking at Table 3, the results are unconvincing, and more explanations should be provided. Generally, adding an appropriate amount of filler could enhance the mechanical properties due to the reinforcing effect. However, from Table 3, adding fillers to the composites

deteriorated the tensile properties of the materials, and this contradicts what was generally reported in literature studies. Please tabulate the mechanical properties of PVC, SF, BF and RH.

13. For the water absorption tests, the authors were supposed to identify the water absorption behavior by plotting the water absorption percentage curves with respect to immersion time. The curves are vital to show the water absorption behavior of composites with different types of fillers and weight compositions.

14. Overall, the results and discussion section needs major revision. The authors just analyzed and discussed the fundamental aspect of each property without providing an in-depth discussion. What is the role of filler in the physical, thermal and mechanical properties of the composites? How did the filler weight composition affect the physical, thermal and mechanical properties of the composites? How did different types of fillers affect the physical, thermal and mechanical properties of the composites?

15. The conclusion needs to be rewritten. It does not reflect the major findings of this work.

Author Response

Answers in the report, attached.

Reviewer 3 Report

Comments and Suggestions for Authors

Journal: Polymers (ISSN 2073-4360)

Manuscript ID: Polymers-2976539

Title: A study of plant-filled polymer composites based on highly plasticized polyvinyl chloride

Authors: Evgenia O. Samuylova * , Alina Ponomareva , Vera Sitnikova , Anton Zhilenkov, Olga Kichigina , M. V. Uspenskaya

Review Comments:

1.      Page No.1, Line No.13, there is a redundant word ‘ study’. Please look into it.

2.      The basis of filler selection and their concentration (% wt.) should be justified in the manuscript.

3.      Page No.2, Lines No. 79, “a powder with a dispersion of 0.2 ÷ 2 mm”. I think it should be ‘– ‘rather than ‘÷ ‘.

4.      More insight must be given in the optical measurements section. Whether the authors observed any agglomeration effects, when the filler % is increased from 20 to 40. If not, under high filler %, how the uniform distribution is  ensured? Elaborate.

5.      In IR spectroscopy, the peaks in Fig. 4(b) must be matched to corresponding chemical group of plant fillers. That will improve the understanding of the readers.

6.      In section 3.4. Mechanical testing, for 60/40-SF sample, the % drop in TS is about 31% while the % elongation is reduced by half, whereas the same trend is not observed with other plant fillers. The authors must substantiate this trend with scientific validations.

7.      Page No. 8, Table 4 name should have been changed.

8.      The section 3.4 concluded that 20% BF is the most suitable filler. But the reviewer wonders why for that composition, the water absorption was not conducted at all?

9.      The authors suggested adding plant fillers would improve the biodegradable characteristics of plasticized polyvinyl chloride. But in the entire manuscript, there is no glimpse of biodegradability test and its results. A section in this aspect would add value to the manuscript.

Author Response

Answers in the report, attached.

Reviewer 4 Report

Comments and Suggestions for Authors

Dear Authors,

The aim of this study was to investigate the structural, thermal, and mechanical properties of PVC-based bio filler composites. The topic is interesting and will be useful in the field. While the topic fits into the journal scope, I have a few concerns regarding the scientific quality of the paper. Please consider revising the manuscript based on the following comments: 

General Comments:

1. Pay close attention to the tone and language throughout the manuscript. Includes but not limited to:

(1) Lines 39-40: “Recently, one of the significant disadvantages … is its long “life span”.” The tone of this sentence is informal. The sentence in Lines 132-133 also has the same issue.

(2) Line 49: a comma is preferred before “but”.

(3) Lines 54-55: “One of the well-known … are wood-polymer …” – “is” should be used instead of “are”.

(4) Lines 209-211: I’d suggest to rewrite the sentence to increase the clarity and readability.

Introduction:

1. I’d suggest authors can provide a more comprehensive literature review on the use of bio fillers in PVC-based composites in the Introduction section. It would also be valuable to compare the current manuscript with other articles in the field to identify any research gaps.

2. Lines 41-46: In lines 41-46, authors introduced the fact that PVC cannot be degraded naturally and mentioned that the addition of bio fillers aims to facilitate its degradation. However, there is a lack of further discussion to support this claim. I’d suggest authors to elaborate more on how bio fillers contribute to the degradation of PVC.

3. Adding a paragraph to introduce the bio fillers used in this study (spruce flour, lignin-free birch flour, and rice husk) is recommended.

4. Please add null hypothesis at the end of this section.

Materials and Methods:

1. I am curious about the rationale behind preparing the specimens in the form of films. Could you please explain the reasons for choosing this format? Additionally, it would be helpful to know if you used any specific method to enhance the bonding between the filler and PVC matrix.

2. I would suggest authors can consider supplementing two tests – (1) SEM analysis on the bio filler-filled PVC and (2) filler density test.

3. Line 79: Please check the symbol used in the phrase "dispersion of 0.2 ÷ 2 mm". Authors can briefly describe the method employed to obtain the dispersion of the ground fiber.

4. Figure 1: In addition to the scheme for the fabrication of plant-filled polymer films, I suggest to add the structural, thermal, and mechanical tests used in the study to the diagram. This would provide a complete overview of the experimental design and enhance the clarity of the diagram.

5. I recommend splitting Section 2.3 into independent sections based on the characterization method used. For example, Section 2.3 could be further divided into 2.3.1 Optical Microscopy, 2.3.2 FTIR, 2.3.3 DSC and TGA, and 2.3.4 Tensile Test. This would allow for more detailed descriptions of each characterization method and enhance the clarity of the manuscript. Additionally, the sample size (n=?) for the tensile test should be included in this section instead of in the Results and Discussion section.

6. In Section 3.5, the authors presented the water absorption test results. The procedures for conducting the water absorption test should be also described in the Materials and Methods section.

7. Check the manufacturer names and locations mentioned in Lines 64, 78, 81, and 82. Both manufacturer name and location should be described in the brackets, i.e., “laboratory rollers (Schwabenthan Maschinenfabrik, Berlin, Germany)”.

8. Line 85: microns à µm; Line 95: cm-1 à cm-1.

Results and Discussion

1. The insert position of Figures 2 and 3 should be changed. All tables and figures should be placed immediately after the paragraph that mentions the specific table/figure.

2. Figures 4 and 5: I recommend changing the marks of the legends (1-7) to actual names, such as Unfilled PVC, 80/20-SF, etc. This would improve the readability of the figures and make it easier for readers to understand the results.

3. Line 188: The authors used student's t-test for multiple comparisons. However, it may be more appropriate to use one-way ANOVA for this analysis. I recommend checking the significance of the tensile test results using one-way ANOVA.

4. Line 201-202: The authors state that “increasing the proportion of plant filler to improve the environmental friendliness of materials is limited by decreasing strength characteristics and flexibility” While I understand that the film specimens are more suitable for tensile tests, it is worth noting that other strength properties, such as flexure and compression strength, are missing from this study. I would suggest that the authors revise this statement by adding “tensile” before “strength” to increase precision.

5. Check the caption of Table 4. Additionally, I noticed that the authors only present results from four groups, instead of all seven groups. Please check and include the results from all groups.

Thank you for considering these comments. I believe that addressing these concerns will strengthen the scientific quality of this manuscript. Good luck!

Comments on the Quality of English Language

The quality of the English language should be improved before publishing. Authors should pay close attention to the tone throughout the manuscript.

Author Response

Answers in the report, attached.

Round 2

Reviewer 2 Report

Comments and Suggestions for Authors

Author Response

Please find the answers inside the file attached.

Reviewer 4 Report

Comments and Suggestions for Authors

Dear Authors,

Thank you for revising the manuscript and the responses. The manuscript has been improved significantly.

I only have three very minor suggestions/comments as below:

1. Lines 98-103: I'd recommend to move the introduction of three bio fillers used in this study to Introduction section, before "The aim of this study..." (Line 82).

2. Line 309: "⁰C" should be written as "°C".

3. Line 318: Delete "i" appeared in the subtitle.

With that, I recommend to accept the manuscript after minor revision. Good work!

Author Response

(The authors gave the same response as above.)

Round 3

Reviewer 2 Report

Comments and Suggestions for Authors

I can see the authors have made substantial revisions to the manuscript based on the comments given by the reviewer. According to the amendment made by the authors, a few minor issues still require the authors to look into them. Please make all necessary revisions before this manuscript can be accepted for publication.

1. On page 5, the authors did not only identify tensile strength but also modulus and elongation. Therefore, the title of section 2.3.5 is inappropriate. Please change the title of section 2.3.5 to ‘Tensile tests’.

2. On page 12; line 313, the sentence ‘In [86-87] was noted that’ should be modified. Generally, it is written as ‘In the previous studies conducted by Fedele and Ata [86] and Li et al. [87], it was noted that…….’

3. The authors identified the water absorption of the composites instead of their swelling behaviors. However, in the manuscript, the authors mention the swelling in several sections, including the abstract, section 2.3.6, section 3.5 and conclusion. Please resolve this issue.

Author Response

Please check the answers in the attached file. 
